# Multisectoral actions in primary health care: A realist synthesis of scoping review

Resham B. Khatri[1,2]*, Daniel Erku[3,4], Aklilu Endalamaw[1,5], Eskinder Wolka[6], Frehiwot Nigatu[6], Anteneh Zewdie[6], Yibeltal Assefa[1]

1 School of Public Health, the University of Queensland, Brisbane, Australia, 2 Health Social Science and Development Research Institute, Kathmandu, Nepal, 3 Centre for Applied Health Economics, School of Medicine, Griffith University, Nathan, Australia, 4 Menzies Health Institute Queensland, Griffith University, Nathan, Australia, 5 College of Medicine and Health Sciences, Bahir Dar University, Bahir Dar, Ethiopia, 6 International Institute for Primary Health Care-Ethiopia, Addis Ababa, Ethiopia

* rkchettri@gmail.com

**Data Availability Statement:** All relevant data are within the paper and its Supporting Information files.

**Funding:** The author(s) received no specific funding for this work.

## Abstract

### Background

Multisectoral actions (MSAs) on health are key to implementation of primary health care (PHC) and achieving the targets of the Sustainable Development Goal 3. However, there is limited understanding and interpretation of how MSAs on health articulate and mediate health outcomes. This realist review explored how MSAs influence on implementing PHC towards universal health coverage (UHC) in the context of multilevel health systems.

### Methods

We reviewed published evidence that reported the MSAs, PHC and UHC. The keywords used in the search strategy were built on these three key concepts. We employed Pawson and Tilley's realist review approach to synthesize data following Realist and Meta-narrative Evidence Syntheses: Evolving Standards publication standards for realist synthesis. We explained findings using a multilevel lens: MSAs at the strategic level (macro-level), coordination and partnerships at the operational level (meso-level) and MSAs employing to modify behaviours and provide services at the local level (micro-level).

### Results

A total of 40 studies were included in the final review. The analysis identified six themes of MSAs contributing to the implementation of PHC towards UHC. At the macro-level, themes included influence on the policy rules and regulations for governance, and health in all policies for collaborative decision makings. The meso-level themes were spillover effects of the non-health sector, and the role of community health organizations on health. Finally, the micro-level themes were community engagement for health services/activities of health promotion and addressing individuals' social determinants of health.

**Competing interests:** The authors have declared that no competing interests exist.

## Conclusion

Multisectoral actions enable policy and actions of other sectors in health involving multiple stakeholders and processes. Multisectoral actions at the macro-level provide strategic policy directions; and operationalise non-health sector policies to mitigate their spillover effects on health at the meso-level. At micro-level, MSAs support service provision and utilisation, and lifestyle and behaviour modification of people leading to equity and universality of health outcomes. Proper functional institutional mechanisms are warranted at all levels of health systems to implement MSAs on health.

## Introduction

Intersectoral coordination is a fundamental principle of primary health care (PHC) depicted in the Declaration on PHC in Alma-Ata in1978. In the Declaration, intersectoral coordination on health was described essential dimension to address the underlying factors of health [1, 2]. This concept includes political, administrative, and technical processes to negotiate and distribute power, resources, and capabilities between and across sectors that improve population health [2, 3]. Alternatively, several terms interchangeably are used to denote intersectoral coordination such as multisectoral actions (MSAs), intersectorality, multisectoral collaboration, or non-health sector interventions. Since Alma-Ata Declaration on PHC, several other global health initiatives also have prioritised the role of MSAs in health, including the Ottawa Charter (1986) [4], the Health in All Policies (HiAP) approach (2007-with Adelaide (2010) [5] and Helsinki (2013) [6] statements), and the Sustainable Development Goals (SDG) 3 (2015) [1, 7]. In the current body of literature, the MSAs is widely used as umbrella term to denote the intersectoral coordination in PHC and contribution of non-health sector interventions. In 2018, Astana Declaration on PHC also reiterated the importance of MSAs and considered it as one of the pillars of PHC towards achieving universal health coverage (UHC) and leaving no one behind [8].

Roles of MSAs in PHC are vital to achieving the health-related SDG and its global targets of UHC. MSAs have the potential to impact population health by influencing both the supply side (e.g., connecting health facilities and communities, contributing to local health governance, planning, and ensuring health services resources) and demand side (e.g., increasing awareness of health care needs) of health and social systems [9]. For example, MSAs are meant to address spillover effects for health through cross-sectoral policies, health sector-led collaborations and coordination, and implementation of non-health sector interventions on health [7].

Moreover, there has been an intricate linkage between MSAs, PHC and UHC. The idea of PHC (with principles of human rights, and community participation), is a means to equity and universality and ensure health services to those populations with unmet health needs and are already left behind [10]. MSAs are inputs for implementing the PHC approach to ensure access to quality essential health services without financial hardship [1, 11]. Thus, MSAs could provide the contexts/inputs and pathways to design and implement PHC services/systems that lead to the delivery and utilisation of PHC services.

The World Health Organisation's (WHO) Social Determinants of Health (SDoH) framework outlines underlying factors beyond health sector, such as structural (social and political) and intermediary (modifiable non health sector factors) determinants [12]. Structural determinants of health include policy and governance actions, economic and social policies, culture, and societal values, influenced by rules and regulations at the higher level. While intermediary

determinants are modifiable factors such as material circumstances (living and working conditions of people and communities), factors affecting populations' behavioural and psychological contexts [13]. Past studies reported several MSAs positively impact community involvement in local governance, resource mobilisation, and delivery of PHC services in hard-to-reach areas and marginalised populations [14–17]. However, there has been limited understanding and interpretation of how actions of non-health sectors articulate their interests, exercise their rights and obligations in policy and practice within health systems and mediate health outcomes [18].

To understand the contribution of MSAs in PHC, a theory-driven and interpretive approach (Pawson and Tilley's realist review) has been used to synthesise evidence on health interventions [19]. Such an approach has the potential to provide comprehensive evidence to guide policy and practice research in the PHC context for better health outcomes. The realist review approach could be an appropriate method to generate findings exploring how MSAs on health operate at multilevel health systems at a strategic level by collaboration and decision-making; contextualisation of policy decisions and plans at an operational level; and implementation for the service delivery level. This study aimed to analyze MSAs on the PHC to answer the following key question: which and how do MSAs contribute to design and implementation of PHC services? This review provides the context of PHC on what works for whom and under what circumstances to deliver PHC services that could guide policymakers in designing responsive policies and actions. Additionally, this review synthesises the contribution of MSAs to PHC systems/services at different levels, including successes and challenges and strategies for implementation. The findings of this study could give new insights/perspectives to health policymakers on different categories of MSAs on health to be designed and implemented in the context of multilevel health systems.

## Methods

We conducted a review of published evidence reporting MSAs in the context of the implementation of PHC towards UHC. We employed Pawson and Tilley's realist evaluation approach [19]. This review was conducted following the Realist and Meta-narrative Evidence Syntheses: Evolving Standards (RAMESES) publication standards for realist synthesis (S1 File) [20].

### Initial scoping of the review question

After preliminary discussion among authors, we developed an conceptual framework showing the interlinkage of MSAs, PHC, and UHC (Fig 1). This framework represented three levels of MSAs influencing the implementation of PHC towards UHC. The MSAs can operate at three levels: a) distal (strategic actions at the higher level), b) intermediary (contextualisation of strategic MSAs at the operational level), and c) proximal (implementation of MSAs to generate health service and delivery and utilisation). This initial program theory further helped to inform the eligibility criteria and develop the search strategy for a systematic search and explaining/interpreting the findings [21].

### Search process

We searched seven electronic databases (PubMed, CINAHL, Scopus, Cochrane Library, EMBASE, PsycINFO, and Google Scholar) for studies that described the contexts, mechanisms, and outcomes of MSAs in PHC. This was followed by complementary searches, including citation searches of included studies, and Google searches to locate further eligible articles that were not identified in the database searches. The keywords used in the search strategy were built on three key concepts and search terms in each concept: MSAs (multisectoralism, intersectorality, coordination, collaboration, multisectoral*, intersectoral*, multisectoral

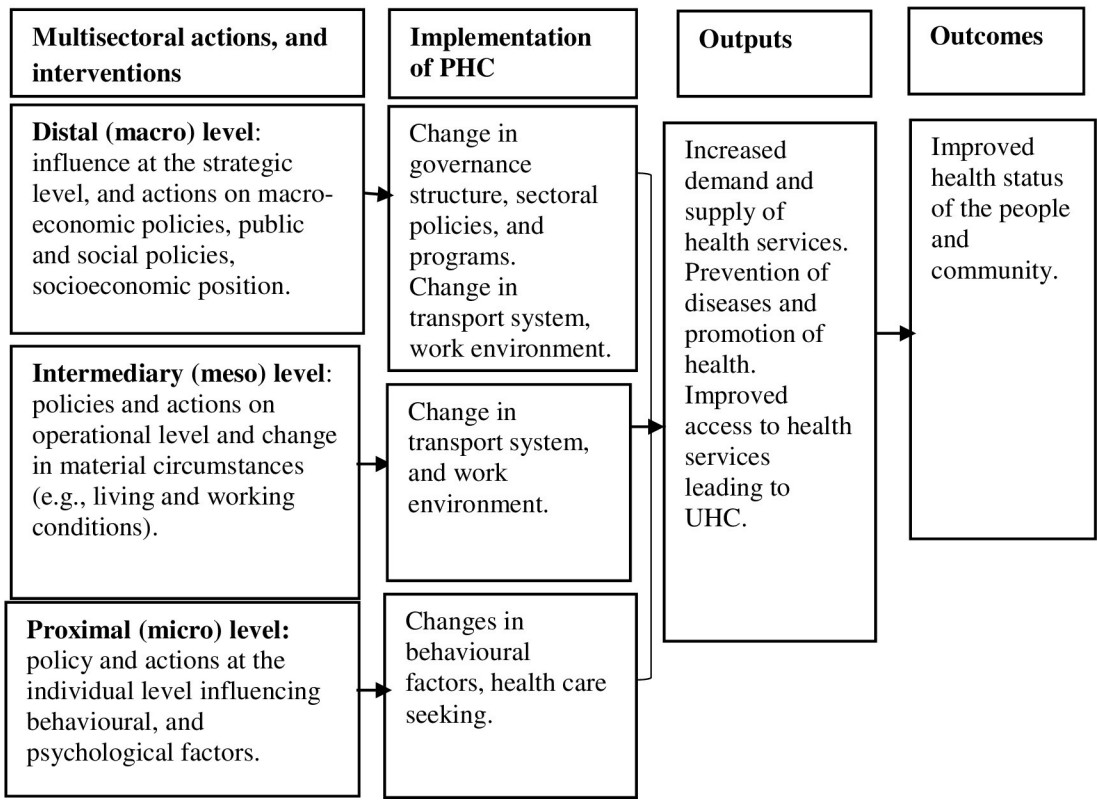

**Fig 1. Initial framework showing interlinkage of MSAs, PHC and UHC.**

action, intersectoral coordination, intersectoral action*); PHC (primary health care); and UHC (universal health care, health services accessibility, quality of health care, safe health care, health coverage, health care coverage, health service coverage, universal coverage, universal health coverage, essential health coverage, health insurance coverage, financial risk protection, financial hardship, financial protection, efficiency, equity, responsiveness, effectiveness, performance). Boolean operators (AND, OR) and truncations (*, " ") were used varied with tailored search terms for each database. The search included articles published in English from the inception of each database until 30 May 2022. No time- or country-related limitations were applied.

## Selection of studies

We included all relevant studies (e.g., quantitative, qualitative, mixed methods, reviews, reports, and secondary data analysis) that were related to the topic of our concern: MSAs in PHC. In addition, we included all studies that informed about MSAs, PHC and UHC. Studies were excluded if information about the underlying mechanisms of MSAs were not explained. Records were managed using Endnote X20 software. Based on the title and abstract, screening was undertaken initially by the first author. This was followed by a full-text screening initially by the second author and assessed by the third author. Any disagreements were resolved by discussion. We took an iterative, holistic approach consistent with the realist synthesis technique based on the relevance of the included studies and was conducted throughout the review process [19]. We considered a study relevant if the data contributed to building the theory after discussion within the review team regarding the methods used to generate data. We

aimed to interpret the findings considering the potential to answer the research question rather than as inclusion criteria and quality of individual study included in the review.

### Data extraction

A data extraction sheet was developed covering author, year, country, types of study, main concept of MSAs, and main findings (S1 File). Data were extracted by the first author and double-checked by the other two authors. Any disagreements were resolved by discussion.

### Analysis and synthesis process

The results were presented as per the RAMESES publication standards for realist synthesis. In addition, the mechanisms that influenced the success and failure of MSAs were synthesised and presented narratively.

## Results

Of 40 studies, there were 21 studies from low-or middle-income countries (LMICs), and nine were from high-income countries (Table 1). A total of six themes were identified (first row, Table 1); of them, two were under macro-level MSAs (blue), two themes under meso-level (yellow), and two themes under micro-level (green) health systems.

Fig 2 presents a selection of studies included in the review. We included 40 studies for the final review (Fig 2).

### Multilevel pathways of MSAs influencing implementation of PHC

This section explains multilevel pathways of how MSAs influence PHC: macro-level actions working as strategic levers, meso-level actions operating to contextualise macro policies and actions, and implementation of MSAs engaging community for health service utilisation (Fig 3).

### Macro-level strategic levers

Macro-level collaborations include strategic levers, describing collaboration's role in driving positive social change and the factors critical to assessing large-scale change initiatives for social innovation, shared value, and collective impact. Marco-level framing considers the MSAs as strategic inputs: policy rules and regulations for MSAs (14 studies), collaborative decision-making and planning (17 studies) at the federal level (S1 File).

**Policy rules and regulations for MSAs.** Macro-level health systems focus on policies, rules, regulations, and resources. However, sectors other than health can contribute to health through governance, policy, research, actions and stakeholders at the higher-level (governments and funders) to promote the goals of the system (policies) [26, 41]. In addition, knowledge and evidence from other disciplines (e.g., political science, development studies, public health) are important for HiAP, plans and programs [26]. For instance, Ethiopia implemented multisectoral policies linked with development (e.g., five-year development plans) and strategies (e.g., poverty reduction strategies) to achieve the targets of global health policies [30]. Similarly, India has employed a robust legal framework, continued engagement of other actors (e.g., policymakers, researchers, advocates, politicians) and actions to address policy incoherence issues on tobacco control initiatives [58]. These macro-level multisectoral policies for tobacco control initiatives have huge potential to control tobacco-related several non-communicable diseases (NCDs).

The intersectoral coordination and functional sectoral institutional mechanisms can potentially reduce social and health-related inequalities in health and nutrition [32, 37, 48, 53, 59].

**Table 1. Summary of studies, country, and themes.**

| Study | Country | Policy rules and regulations | Collaborative decision making and planning | Spillover effects | Community health organisations | Community engagement | Health promotion and prevention |
|---|---|---|---|---|---|---|---|
| Ramírez et al. [22] | South America | | | | x | x | |
| Labonté et al. [23] | LMICs | | | | | x | x |
| Álvarez-Bueno et al. [24] | Multiple countries | | | | | | |
| De Andrade et al. [25] | Latin America | | | x | | | |
| Evelyne de Leeuw [26] | Multiple countries | x | | x | | x | |
| Fisher et al. [15] | Australia | x | x | | | | x |
| van Eyk et al. [27] | Australia | x | x | | | | |
| Maluka et al. [28] | Tanzania | | | x | | | |
| Javanparast et al. [29] | Australia | | | | x | x | |
| Assefa et al. [30] | Ethiopia | x | | | | | |
| Jimenez et al. [31] | El Salvador | | | | | x | x |
| Kraef et al. [32] | Uganda | x | x | x | | | x |
| Kriegel et al. [33] | Austria | | | | | x | |
| Tumusiime et al. [34] | African countries | x | x | x | x | | |
| Bermejo et al. [35] | Cuba | | x | | | | x |
| DeHaven et al. [36] | USA | | | | | x | |
| Feryn et al. [37] | Multiple countries | x | x | x | | x | |
| Hazazi et al. [38] | Saudi Arabia | | | x | | | |
| Nolan-Isles et al. [39] | Australia | | | | x | x | |
| Sitienei et al. [40] | Kenya | | | | | x | |
| Sturmberg et al. [41] | Multiple countries | x | | x | x | | |
| Super et al. [42] | Netherlands | | x | | x | | |
| Tuangratananon et al. [43] | Thailand | | | | | x | x |
| Madon et al. [44] | India | | | | | x | |
| Perveen et al. [45] | Multiple | | | | | x | x |
| Rahimi et al. [46] | Iran | | x | | | | |
| Holveck et al. [47] | South America | | x | | | | |
| Adeleye et al. [48] | Multiple | x | x | | | | |
| Shankardass et al. [49] | Multiple | | x | | | | |
| Spiegel et al. [50] | Cuba | | | | x | x | |
| Ndumbe-Eyoh et al. [51] | Multiple countries | | x | x | | | x |
| Rudolph et al. [14] | Multiple countries | | x | | | | |
| Anaf et al. [52] | Australia | x | | x | | | x |
| Souza et al. [53] | Brazil | x | x | x | | x | |
| Souza et al. [54] | South America | | x | | | | x |
| Chaudhary et al. [55] | Nepal | | | | | | |
| Dhimal et al. [56] | Nepal | | x | | | | |

(*Continued*)

**Table 1.** (Continued)

| Study | Country | Policy rules and regulations | Collaborative decision making and planning | Spillover effects | Community health organisations | Community engagement | Health promotion and prevention |
|---|---|---|---|---|---|---|---|
| Ruducha et al. [57] | Nepal | | x | | | x | x |
| Mondal et al. [58] | India | x | | | | | |
| Salunke [59] | India | x | | | | | |

Intersectoral collaboration can contribute to design resilient health systems through advocacy, sectoral integration, and mobilisation of multisector stakeholders [25, 34, 48]. Nevertheless, macro-level MSAs were constrained by resource limitations, political priority and policy context,

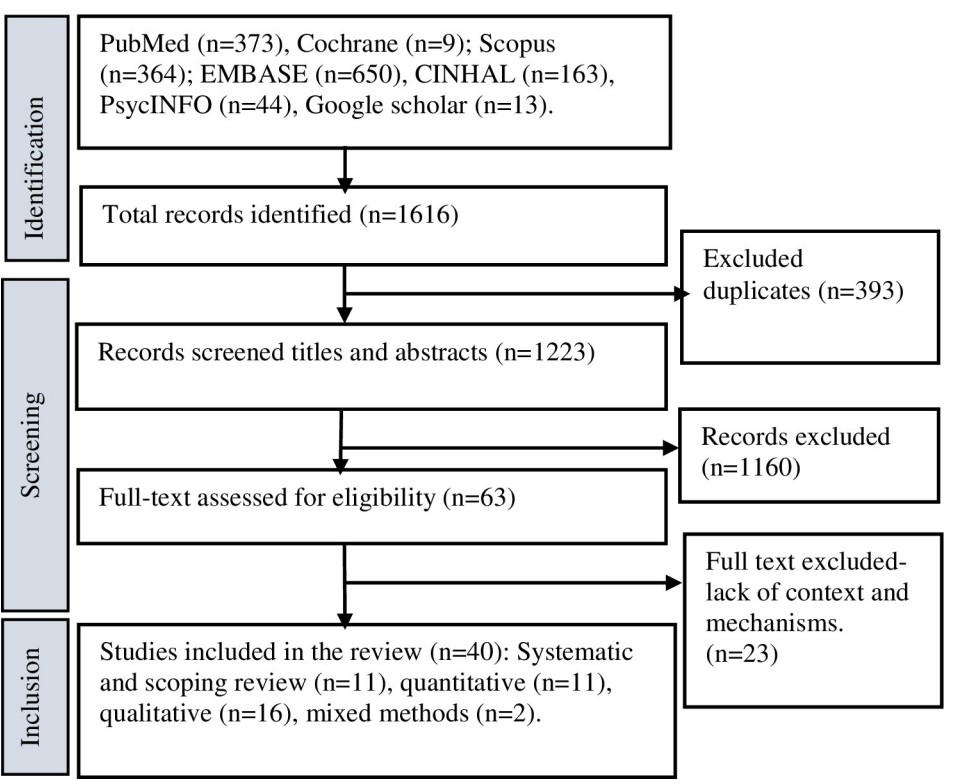

**Fig 2. Selection of studies for realist synthesis.** Adapted from [20].

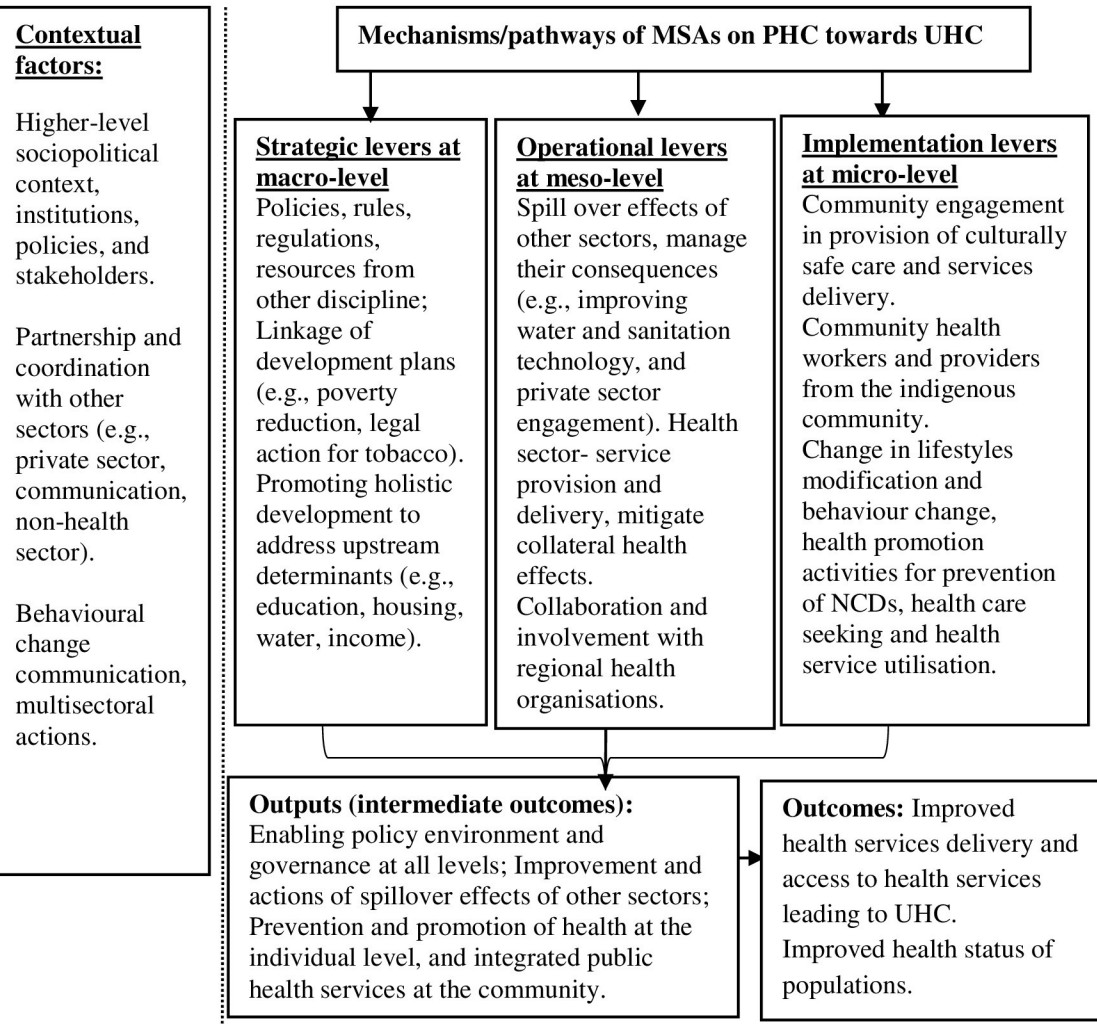

**Fig 3. Pathways/mechanisms of MSAs that influence PHC towards UHC.**

and system (e.g., rules, regulations, resources), including poor understanding of health in all policies as tools for improving the process of intersectoral policy development [15, 27, 41, 52].

**Collaborative decision-making for planning in health in all policies.** Intersectoral collaboration can work as the strategic inputs/levers to tackle the upstream social determinants of health (e.g., poverty). Primarily, impact health outcomes are shaped by decisions outside the health sector, including changes in government organizational structures and processes [14, 48]. Such non-health sector policy decisions influence on the SDoH, while the health sector expects inputs from this sector with shared responsibility for health improvement [14, 48]. Improving health outcomes need to address upstream determinants of health (e.g., poverty, income), which require the activities of the health sector and sectors that influence health (e.g., education, housing, water and sanitation, labor, public works, transportation, agriculture, and social justice-based approach) [32, 37, 42, 47, 51, 53, 54].

Approaches of HiAP incorporate collaborative decision-making, facilitates joined-up process working with key actors with power (e.g., the ministry of finance), and covers health considerations across sectors and policy areas, contributing towards co-benefits of the sectors along with addressing SDoH and inequities [14, 27, 34]. There were neglected tropical diseases

(NTDs), and malnutrition have a vicious cycle of poverty that led to poor health outcomes especially in most poverty-stricken communities [47]. These intricately linked relationships of poverty and poor health demonstrate the importance of linking the health sector's activities with those of other sectors [47]. Several NCDs (such as tobacco-induced NCDs) and injury-induced morbidities and mortalities (e.g., road traffic accidents, trauma) can be prevented through MSAs and incorporating the HiAP approach and strengthening PHC systems [15, 32, 46]. Nepal's Multisectoral Nutrition Plan (MSNP) is an example of HiAP where MSAs occurred in policy dialogue, strategic planning and implementation [57]. Similarly, not only in NCDs and nutritional issues, but macro-level also MSAs were effective in responding to the pandemic. For example, Cuba implemented HiAP approach in a single national intersectoral government plan for COVID-19 and guaranteed effective pandemic management [35]. None-theless, challenges of HiAP included poor understating, superficial and sometimes absent MSAs, lack of leadership and poor dissemination of MSAs (especially in NCDs), poor distal level outcome-focused, instability in bureaucratic leaders, and lack of proper mechanisms of institutionalisation [27, 49, 56]. Thus, the focus of HiAP should be on the distribution of resources and policy settings to improve the socioeconomic status of the populations [15].

## Meso-level operational levers of MSAs

The meso-level lens of MSAs incorporates partnerships and coordination with other sectors (S1 File). Two themes were identified at this level: spillover effects of the non-health sector (12 studies) and the role of regional health organisations (8 studies). Macro-level MSAs could influence to operationalise and make decisions at the provincial level.

**Spillover effects of the non-health sector.** Many health issues that ministries deal with are 'spillovers' or collateral effects from other sectors (e.g., water and sanitation, communication, transportation). Only health system efforts cannot address health sector problems as they are socially constructed and hierarchically layered organisational systems [41]. Social participation, MSAs between public health and other sectors, and awareness of the social justice-based approach create enabling environments and enhance access to services targeting marginalised populations [25, 26, 32, 37, 51, 53]. Partnerships with private sector and digital technology could support creating an enabling environment to address other sectors' collateral impacts on the health sector. Private sectors' engagement and digital technology can help to provide information, documentation, and health services [28, 38]. Evidence suggested that mobilisation of other sectors was effectively adopted in the earthquake response in Nepal [55] and the cholera epidemic response in many African contexts [34].

While addressing collateral effects non health sector, several factors influenced the health sector (e.g., human resources with diverse backgrounds and motivations, limited financial resources and delayed reimbursements, weak administrative functions and technical capacity of local authorities, and lack of trust between the governments and private partners) [28, 52].

**Community health organisations for MSAs.** Health organisations could operationalise the MSAs. For example, regional organisations (e.g., Medicare Locals/Primary Health Networks), Aboriginal Community Controlled Health Organisations, and local health committees can interact with community organisations and policymakers' provision of health services (availability, reliability and affordable) to serve marginalised communities [29, 39]. In addition, these organisations enhance better governance for horizontal coordination, and demand accountability to ensure the capture of political commitment and social innovation for improvements [42, 44]. Systematic and regular engagement of multisector and agencies contributed to integration, knowledge exchange and implementation, resilience, and antifragility [34, 50]. Using a participatory bottom-up approach, local agencies identified, prioritised, and

formulated local intersectoral public health policy and plan to maintain health system goals for improved health services targeting marginalised communities [22, 41]. Thus, organisational capacity, resources, governance system and legislation are critical for collaborative planning at the regional level [29]. Nonetheless, inadequate coordination between local and national interests influenced the operationalisation of MSAs on health [22].

## Micro-level implementation levers: Health promotion and integrated health services

Two themes were identified at this level: community engagement for service delivery (16 studies), and health promotion and preventive activities (12 studies). The micro-level lens of MSAs in PHC describes to integrate efforts in public health: assessment and planning at the local level, implementing targeted measures, changing conditions in communities and systems. Multisectoral actions at this level aim to implement programs for service delivery and utilisation, and change in individual-level factors (e.g., modification of behaviours, lifestyle change) towards improving health and health equity (S1 File).

**Community engagement for service delivery.** Participation with the community is important for MSAs and addressing the SDoH of people and communities. Regular and systematic community engagement of different sectors, other agencies, and civil society and community perspectives can co-develop healthcare solutions across the broad range of causal factors in addressing health determinants [26, 36, 50]. Community engagement in PHC promotes communication between service providers and users, cultural safety and acceptability, and appropriateness of health programs [39, 45]. For example, women's groups supported seed funding and developed insurance funds to assist pregnant women in the Democratic Republic of Congo [23]. Community participation, intersectoral actions in local planning and working directly with and in affected communities in high-risk communities increase the involvement of citizens, and community empowerment of local communities for the utilisation of the services to marginalised populations [22, 26, 36].

Social workers improved PHC services using their professional competencies, cooperation and communication with stakeholders, as social workers had broad perspectives and integration into PHC [33, 37]. Health workers from indigenous communities also increased awareness of pregnancy and childbirth health and the provision of health services among disadvantaged populations [23, 39].

Community engagement and communication supported implementing MSAs, engaging village health volunteers to raise awareness of NCDs, supporting screening enrolment and adhering to interventions [43]. Community engagement through village health volunteers improves malnutrition awareness, supports enrolment in screening and raises adherence to interventions [57]. Mobilisation of CHWs and interdisciplinary teams contributed to the provision of need-based, equitable and intercultural practices of maternal and child health services (e.g., immunisation, family planning, HIV program) in hard-to-reach areas [22, 23, 33]. The CHWs brought holistic health services closer to the communities and hard-to-reach regions of Latin America (e.g., El Salvador) [22, 31]. Village Health Sanitation and Nutrition Committees in India focused on sanitation, nutrition, and hygiene, which impede improving PHC amongst poor and marginalised communities [44]. The committee members with retired government workers had political connections, power, and influence in the decision-making [40].

However, challenges of MSAs of community-based organisation and committees were poor participation, duplication, lack of clarity of responsibilities, limited time and financial support, limited collaboration with the local government, lack of inclusion of marginalised groups, unclear process of involvement and conflict of interests, and variable competence of

committee members [29, 40]. Implementing MSAs locally was hindered by inadequate community participation and coordination, poor communication and articulation among sectors for integrated planning, and individualised social interventions [22, 37, 53].

**Addressing downstream social determinants of health.** MSAs at the micro-level (service delivery) can prevent NCDs and NTDs and promote people's health by linking the community and health promotion activities. In addition, micro-level MSAs can potentially address the downstream SDoH of individuals by adopting a healthy lifestyle and implementing interventions outside the health sector [51, 54]. Moreover, health literacy empowers individuals and citizens to optimize their health, linking community and health system to prevent and control NCDs and malnutrition [43, 57].

Most NCDs are the product of the dynamic of urbanisation and socialisation that originate from interactions of obesogenic environments, urbanisations, and lifestyles requiring multifactorial actions to address the risk and prevention of diseases [43]. The El Salvadoran health system emphasized PHC systems that brought holistic care closer to the communities to prevent NCDs using interdisciplinary teams and health promoters [31]. Multifactorial community interventions were effective potential risk factors of cardiovascular diseases among high-risk populations (e.g., controlling blood pressure and cholesterol level) [24]. Addressing malnutrition (undernutrition and obesity) requires multisectoral nutrition-sensitive interventions (e.g., community empowerment for healthy diets) and nutrition-specific interventions (e.g., treatment of undernutrition) [32]. Moreover, Kenya's community health strategy included safe access to safe drinking for the improved health status of mothers and children [23]. Cuba's response to the COVID-19 pandemic included a range of actions, such as preventive measures in the community, continued isolation centres, and community with actions of surveillance and follow-up of recovered patients [35]. However, MSAs were lacking in health programs, while health services were individualised medical or behavioural interventions focusing on proximal factors for client groups with extending access to health services [15, 45, 52].

## Discussion

Using a realist synthesis approach, this study synthesised available evidence on the MSAs on implementation of PHC in multilevel health systems. Three mechanisms of MSAs contributed to policy and implementation of PHC: working as strategic, operational and implementation levers. First, strategic macro-level actions contributed to all policy collaboration in governance, and health actions; second, the meso-level MSAs effectively addressed the spillover effects of other sectors by contextualising strategic levers. Third, micro-level MSAs were found to be engaging the community and services providers for health promotion, prevention of diseases and provision of integrated public health services.

It is vital to realise the importance of MSAs, which can be improved through following mechanisms. First, macro-level mechanisms include collaboration of sectors and change in governance and policy, legal and monitoring framework addressing social determinants of health. At the macro-level, MSAs include the strategic actions of cross sectors for integrated planning and actions on health(e.g., prevention of NCDs and malnutrition or alcohol and tobacco control) [60]. One example of strategic MSAs is Nepal's multisectoral action plan on NCDs that accelerates and scales up the national multisectoral response to NCD-epidemic by setting functional mechanisms for multisectoral partnerships and effective coordination, effective leadership and sustained political commitment and resources for its implementation [61]. Additionally, during the COVID-19 pandemic, many countries adopted HiAP to contain the pandemic by collaborating with other sectors (e.g., border protection, economic support, communication, and education). In addition, they made institutional arrangements (e.g.,

secretariat, strategic committee, working group) at the federal level [62]. The interest of stakeholders, with shared ideas and institutions, could work for joint planning and produce policy and strategic documents. There are case studies of the macro-level intersectoral actions, including led agency actions to reduce road traffic accidents in Iran [63], tobacco control policies and actions in India [58], and MSNP for improving nutritional outcomes in Nepal [57]. In Indonesia, there was irrational use of antimicrobial resistance in formal health care facilities, communities, and beyond health sector such as the livestock and fishery sectors, and farms; but lacked multisectoral coordination between sectors [64]. Nevertheless, macro-level collaboration challenges include leadership of the MSAs and accountability and reporting mechanisms.

Second, the contribution of MSAs has spillover effects of the non-health sector and contributes to health systems and services. Higher level policy and actions require meso-level actions creating an environment for contextualising macro-policies. For example, any disaster, earthquake, or climate change can impact multiple sectors with collateral effects (e.g., road, transport, communication, water sanitation system) [55]. These effects can have collateral effects on health (e.g., water and sewage system damage can increase water-borne diseases, poor road networks hinder access to health facilities and health services, shortage of food supply and undernutrition) [65]. Other sectors can fix these multiple effects using the emerging sector. For instance, information systems and digital technology can bridge the gap between communities and service centres [66].

Similarly, coordination with the private sector can be instrumental in health care and sectoral service delivery working with the public sector [59]. Fixing other sectors' effects can reduce the health system's burden, which requires proper communication, the joint meeting of sectors, and sharing work plans and activities of the cross-sector influencing the health sector [67]. Other sectors play their roles, while the health sector leads in sharing and informing health impacts and coordinates for the multisectoral response mechanisms [68].

Third, micro-level MSAs are vital for service delivery (person-centred and population services- behavioural modification, prevention, and promotion of NCDs) [69, 70]. For this, health literacy and awareness have huge potential to address the impacts of commercial determinants of health, empowering individuals and citizens [43, 71]. Additionally, services provided by an interdisciplinary team and CHWs reaching the community are examples of micro-level MSAs integrated with public health services [43]. Implementing population health services requires community engagement and participation, encouraging people to seek care. Community participation and community health workers are linked to strengthened primary-care facilities and first-referral services incorporating health and development for better water, sanitation, nutrition, food security, and chronic diseases [72, 73].

Some of the MSAs intersect and operate at the multilevel of health systems. For example, a substantial group of commercial actors are escalating avoidable factors of ill health, planetary damage, and inequity due to products and practices that greatly cost individuals and society [74–76]. Commercial determinants such as products (e.g., goods and services) that are not unhealthy commodities affect at the micro-level, especially individual behaviours, and lifestyle influencing human health and illnesses [76]. Commercial actors can escalate harms, increase costs by impoverishing and disempowering, and leave health-care systems unable to cope with those multilevel impacts [75]. While MSAs such as progressive economic models, frameworks, government regulation, compliance mechanisms for commercial entities and models incorporate social and health, goals, and strategic civil society mobilisation offer possibilities of systemic change that have the potential to reduce those harms arising from commercial forces, and foster human health wellbeing [74].

The contribution of MSAs is undermined in the context of dominated economic policies and government priorities. For example, many countries establish tobacco or alcohol industries and generate taxes, while their health systems focus on establishing cancer hospitals rather than investing in tobacco prevention and control programs. Additionally, poor design and implementation of MSAs lies on the root causes of the challenges such as poor governance, including entrenched political and administrative corruption, widespread clientelism, lack of citizen's voice, weak social capital, and lack of trust and respect for human rights. This is further complicated by the lack of government effectiveness due to poor capacity for strong public financial management and low levels of transparency and accountability [77]. MSAs are neglected in three prongs: perception of 'non-health' PHC strategies are outside the statutory control of the health sector; lack of practical initiatives from the health sector towards intersectoral collaboration; and PHC is not being the agenda of 'non-health' sectors [78]. Financial sustainability is a challenge for MSAs in many LMICs as those countries' health systems depend on donor aid from high-income countries. Moreover, frequent administrative leadership changes also influence the implementation of MSAs.

Multisectoral actions can contribute to PHC by enabling other sectors' policy actions influencing the health sector. In many resource-limited settings, stakeholders prefer to implement the quick fix approaches (knot and bolt approach) for immediate results, while addressing SDoH and achieving sustainable and long-term health outcomes require holistic multisectoral approaches [79]. Thus, understanding MSAs should move away from the lens of the health system (services delivery and utilisation) to leverage actions of other sectors at the strategic and operational levels.

This study has some strengths and limitations. We conducted a narrative review using theory-driven methodology synthesising data and interpreting findings using a multilevel lens. We could not follow all steps of the realist review approach (e.g., testing of theory and stakeholders' engagement. We broadly followed the RAMESES checklist to select studies, framed/theorised review findings using a multilevel framework, and provided a few case studies (using context, mechanisms, and outcomes configuration). We reviewed the roles of MSAs on PHC by exploring actions at the strategic, operational and implementation levels that directly or indirectly contribute towards UHC. As UHC has three dimensions- notably- coverage of population, service and financial risk protection, we could not focus on each component of UHC. Rather we looked at the UHC through the equity lens. Future studies can be conducted focusing on MSAs' role in specific UHC components. Finally, we synthesised the findings using available literature; future studies can be conducted by conducting stakeholders with rich experience in MSAs in PHC systems.

## Conclusion

Multisectoral actions can contribute to PHC by enabling policy actions of other sectors and a wide range of stakeholders involved in the processes influencing at macro, meso and micro levels of health systems. Macro-level MSAs ensure strategic policy actions on health by bringing actors and sectors together. Higher-level working groups, coordination committees, and multisectoral secretariat are some macro-level institutional arrangements that could collaborate for MSAs on health. Such arrangements could help to produce multisectoral policies, plans and strategies to address macro-level SDoH. At the meso-level, stakeholders of the non-health sector could fix the spillover effects of health, potentially reducing the collateral impacts on health systems. At the micro-level (individual level and local service delivery outlets), MSAs focus on lifestyle modification and behaviour interventions, including health literacy, awareness and empowerment of individuals and citizens to optimize their health. In the multilevel

context of health systems, MSAs contribute to PHC through strategic collaboration at the macro-level. At the same time, contextualisation of macro-policy and decisions at the meso-level addresses the collateral impacts of non-health sector on health systems/ and services. Multisectoral policies and actions at these levels provide the contexts for the implementation of PHC. At the micro- or service delivery level, MSAs support service provision and utilisation (supply side), and modification of lifestyle and behaviour change of people (demand side), leading towards equity and universality of health services.

## Supporting information

**S1 File.**
(DOCX)

## Author Contributions

**Conceptualization:** Resham B. Khatri, Daniel Erku, Aklilu Endalamaw, Yibeltal Assefa.

**Data curation:** Resham B. Khatri, Aklilu Endalamaw, Frehiwot Nigatu, Anteneh Zewdie, Yibeltal Assefa.

**Formal analysis:** Resham B. Khatri, Daniel Erku, Eskinder Wolka, Anteneh Zewdie, Yibeltal Assefa.

**Investigation:** Resham B. Khatri, Frehiwot Nigatu, Anteneh Zewdie, Yibeltal Assefa.

**Methodology:** Resham B. Khatri, Daniel Erku, Aklilu Endalamaw, Eskinder Wolka, Yibeltal Assefa.

**Project administration:** Frehiwot Nigatu.

**Resources:** Resham B. Khatri, Eskinder Wolka, Frehiwot Nigatu, Anteneh Zewdie, Yibeltal Assefa.

**Software:** Resham B. Khatri, Daniel Erku.

**Supervision:** Yibeltal Assefa.

**Validation:** Resham B. Khatri, Daniel Erku, Aklilu Endalamaw, Eskinder Wolka, Frehiwot Nigatu, Anteneh Zewdie, Yibeltal Assefa.

**Visualization:** Resham B. Khatri, Aklilu Endalamaw, Yibeltal Assefa.

**Writing – original draft:** Resham B. Khatri, Aklilu Endalamaw.

**Writing – review & editing:** Resham B. Khatri, Daniel Erku, Aklilu Endalamaw, Eskinder Wolka, Frehiwot Nigatu, Anteneh Zewdie, Yibeltal Assefa.

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
