## [Decision Letter · Decision Letter 0]

29 May 2023

PONE-D-23-09012Multilevel multisectoral policy and actions in primary health care: a realist synthesis of scoping reviewPLOS ONE

Dear Dr. Khatri,

Thank you for submitting your manuscript to PLOS ONE. After careful consideration, we feel that it has merit but does not fully meet PLOS ONE’s publication criteria as it currently stands. Therefore, we invite you to submit a revised version of the manuscript that addresses the points raised during the review process.

 Some of the revisions requested in the comments below on editor's additional comments for e.g., that of introduction, methods and ensuring the match of discussion and conclusion with the objective would be required for moving ahead accepting this manuscript. And there are several other issues noted by the editor and the reviewers which require a through revision of the manuscript.

We look forward to receiving your revised manuscript.

Kind regards,

Krishna Kumar Aryal

Academic Editor

PLOS ONE

Journal Requirements:

Additional Editor Comments:

A very good effort was made by authors Resham Khatri et al for this scoping review on multisectoral actions on health. You have tried to bring in several aspects

Introduction

The intro seems to be giving scattered definitions of multisectoral action without enough coherency. The authors could start with one definition that they think is the best and then after that link the other points or even definitions that you want to use in a coherent way linking with each other, such that it gives readers a nice read. And people can follow what the authors are aiming for. In addition, there is a lot of redundancy. For e.g. the line about Astana declaration is mentioned twice in first and third para, points about alma ata declaration has similar presentation.

The introduction has to be thoroughly reviewed, and even may be shortened to make it concise and give a clear message of what exactly is this review about. As said above it has so much information in it, but a clear linkage and a flow is seriously missing, when you come down to your objectives. Even at the end of introduction where you say that this review explores the multisectoral approaches to health…….. I am struggling to see it clearly matching with your research question.

Methods: I was expecting to read or see a framework of how you are trying to link multisectoral actions with PHC and UHC. And then describe what kind of relationships or effects on each other you are trying to demonstrate through this review. I would also like to see what components of PHC you are trying to link in this review. And similarly, what dimensions of UHC in the UHC cube are you trying to link in this review. And what would you ultimately expect to show by this review. All these seem completely missing.

Results

Line 248-249: what do you mean to say by many NTDs-related morbidities cause tobacco and road traffic accidents. And the ref you have cited for this statement (21, 28, 36) do not seem relevant to this line.

Line 278: in one of the other errors of wrong (?) citation where you cite an information about earthquake response in Nepal from a publication from southern Australia. Please check these kinds of errors. There can be many which might go under radar of editor or reviewers. By taking a complete responsibility of ensuring a valid write up with proper citation, I suggest a complete run-through and thorough review of all citations and correcting any errors that the paper contains. This is also highlighted by one of the reviewers.

Line 285 and para following it. You talk about community health organizations and have mixed that with committees for MSAs. However, the community level activities fall under micro level levers as per your framework. I suggest not mixing up the community engagement even if it is through participation in the local health facility committees. Please delineate that from the meso level committees and put it in micro level.

Reaching towards the end of the results section, I was expecting some key findings on how you show linkages of multisectoral actions PHC and UHC. I could see something on PHC but found that UHC is completely missing. I struggled to note what exactly you found about UHC and which dimension of UHC is supported with this evidence.

Discussion and conclusions: also, as commented by one of the reviewers, I struggled to pick a take home message from this review. I would say a lot of information and some limitations as commented above have made you struggle to come up with a clear conclusion. The objective and the research question were not found to have been supported with the conclusion.

Minor comments: The language part needs to be thoroughly checked. There are issues like inconsistent use of capital letters in a number of places. And the issues pointed out by one of the reviewers on use of abbreviated forms also has rampantly occurred in the paper. It looks like the authors have not given enough attention to review their final version of the paper. Please correct all the language issues including these.

At the end, I would suggest that the authors take a serious look on their aim of what they expected to depict from this review, revise the methods section as mentioned above and also other sections. The paper in the current form does not match the journal’s standard without a major revision.

Thank you!

Reviewers' comments:

Reviewer's Responses to Questions

**Comments to the Author**

1. Is the manuscript technically sound, and do the data support the conclusions?

Reviewer #1: Partly

Reviewer #2: Yes

2. Has the statistical analysis been performed appropriately and rigorously? 

Reviewer #1: N/A

Reviewer #2: N/A

3. Have the authors made all data underlying the findings in their manuscript fully available?

Reviewer #1: Yes

Reviewer #2: Yes

4. Is the manuscript presented in an intelligible fashion and written in standard English?

Reviewer #1: Yes

Reviewer #2: Yes

5. Review Comments to the Author

Reviewer #1: I congratulate the authors for concluding their scientific efforts logically, but I would recommend them to consider the following aspects while finalizing the manuscript.

1. Title of the manuscript: I don't think the word "multilevel" is required in the title. I would suggest removing this word from the title. As Principles of PHC has spelled- "Multi-sectoral Coordination" not the "Multi-sectoral action". I am afraid, Multi-sectoral action may misguide the readers.

2. The authors are presenting the findings which are more relevant to LMICs/Limited resources settings which may mislead the readers that 'multi-sectoral coordination' (? Actions) is not relevant in high income countries/resource abundant settings.

3. While reviewing the manuscript, I tried to pick a take home message that could be useful for health system workers/decision makers, but I found no new idea/message from the manuscript. Multi-sectoral coordination had been identified as one of the core principles of PHC since 1978. What new idea/knowledge this study adds in the PHC science? Did the authors just attempt to use the RAMESES standards/check list in the area of PHC. I would suggest authors to highlight on the key but nobel idea/knowledge.

Reviewer #2: Dear Authors

Thank you for your hard work and intelligent piece of the write up to bring forward theoretically guided scoping review for the multilevel and multi-sectoral actions in primary health care. The interesting finding to highlight the spillover effect of the non-health sector actors is appreciated.

Please find some feedback for your consideration:

- Introduction: Line 66 better to mention1978; can combine line 69 and 73; line 81-86 secondary citation is used, better to quote primary references

- The search strategy includes Universal Health Coverage, the results and discussion section do not much highlight on the inter-relation of the delivery of PHC to achieve UHC; fails to give clear picture to the reader. This linkage is also missing in the introduction part. Justification added to why UHC was used in the search strategy and linking it up to the rest of the article is needed. Suggested title "Multilevel multi-sectoral policy and actions in primary health care for achieving universal health coverage: a realist synthesis of scoping review"

- Exclusion criteria are not clear. Line 161/162 reads "sufficient information"; what do this mean is not clear, please mention concretely what it meant to the authors so as to bring readers on the same page and add these to the flow diagram as reasons of exclusion

-The citation though has article on multi-sectoral actions for NCDs (64), very less is described in the result and discussion on aspects of macro and meso level from NCDs lens; case studies on Multi-sectoral action plan for prevention and control of NCDs for any country (example:https://www.who.int/docs/default-source/nepal-documents/multisectoral-action-plan-for-prevention-and-control-of-ncds-(2014-2020).pdf?sfvrsn=c3fa147c_4 ) is suggested. In addition AMR is a known multi-sectoral agenda; suggestion is to review on case study on AMR for the spillover effect (example: https://www.hindawi.com/journals/jtm/2022/2783300/). These literatures will work to strengthen the comprehensiveness of the review and the conclusion.

- In supplementary table, health literacy has come up as one of the pertinent finding; it is one of the important factors acknowledged for lifestyle changes and behavior modification; suggested to bring health literacy to the study results, discussion and conclusion too.

- Commercial determinants of health has been kept in the supplementary table, however it has not been taken up in the main write up. Suggestion to include it to integrate the list in the discussion that is fitting into commercial determinants of health (https://www.thelancet.com/series/commercial-determinants-health)

- Since it is scoping review, it will have its certain limitations: Add limitation of your study

Language impression: There are instances of use of the short forms without their full forms and inconsistent short forms used; eg- Line 91 UHC, Line 235 SDH- SoDH else where; Line 355 NCDs and so on. Suggestion to use short form followed by full form for the first time use and then consistent to use the same short form thereafter.

Thank you for the excellent work.

Best Wishes

A

6. PLOS authors have the option to publish the peer review history of their article (what does this mean?). If published, this will include your full peer review and any attached files.

Reviewer #1: **Yes: **Bhim Prasad Sapkota

Reviewer #2: **Yes: **Ambika Thapa Pachya

---

## [Author Response · Author response to Decision Letter 0]

9 Jul 2023

Point by point to the editor's and reviewers’ comments 

The authors team would like to thank both reviewers and editor for their in-depth review and feedback on our manuscript. We agree with your views and comments, therefore, we revised as suggested. We appreciate your insightful and constructive feedback. In this document, we have responded point-by-point responses on your feedback, and clarification of the concerns where necessary. 

Additional Editor Comments:

A very good effort was made by authors Resham Khatri et al for this scoping review on multisectoral actions on health. You have tried to bring in several aspects.

Response: We thank the editor for your thorough review and feedback on our work. We have incorporated all feedback in the revision. 

Introduction

The intro seems to be giving scattered definitions of multisectoral action without enough coherency. The authors could start with one definition that they think is the best and then after that link the other points or even definitions that you want to use in a coherent way linking with each other, such that it gives readers a nice read. And people can follow what the authors are aiming for. In addition, there is a lot of redundancy. For e.g. the line about Astana declaration is mentioned twice in first and third para, points about alma ata declaration has similar presentation. The introduction has to be thoroughly reviewed, and even may be shortened to make it concise and give a clear message of what exactly is this review about. As said above it has so much information in it, but a clear linkage and a flow is seriously missing, when you come down to your objectives. Even at the end of introduction where you say that this review explores the multisectoral approaches to health…….. I am struggling to see it clearly matching with your research question.

Response: thank you, editor, for this insightful feedback. In the introduction section, we started conceptualizing multisectoral actions with some chronological background, followed by the role of MSAs on health, linkage of MSAs, PHC and UHC. Furthermore, we also provide the context of the linkage of MSAs, PHC and UHC- framing MSAs (as inputs) to PHC (as a process) towards universal health coverage (as final outcome of health systems). The main objective of this review is to synthesize how MSAs (inputs/mechanisms or pathways) support the design and implementation of PHC at multiple levels of health systems. As suggested, this section is shortened.

Methods: I was expecting to read or see a framework of how you are trying to link multisectoral actions with PHC and UHC. And then describe what kind of relationships or effects on each other you are trying to demonstrate through this review. I would also like to see what components of PHC you are trying to link in this review. And similarly, what dimensions of UHC in the UHC cube are you trying to link in this review. And what would you ultimately expect to show by this review. All these seem completely missing.

Response: as this study is a literature review (scoping review particularly), we aim to synthesize the evidence of how MSAs at the multilevel health systems influence PHC towards health equity (in particular UHC) (figure 1). The conceptual interlinkage of MSAs, PHC and UHC has been described in the introduction section. Our literature search approach is conceptualized under three themes (MSAs as inputs in the system), PHC as implementation approach, and health equity (UHC) as an outcome. Under the UHC includes search terms based on health system goals (equity, efficiency, access, coverage). Multisectoral actions in PHC are equally important in all three dimensions of UHC (service, population, and financial coverage), therefore, we used the umbrella term UHC rather than a specific component of UHC. Our focus of inquiry is how MSAs contribute to the design and implementation of PHC at the multiple that lead (directly and indirectly) towards health equity. As our RQ question in the research was how MSAs influence PHC in the context of multilevel health systems, we framed our data synthesis framework as macro health systems as policy level MSAs, spillover health effects at the meso level health systems, and influence of MSAs in the service delivery. In this review, we aimed to se evidence on the MSAs and PHC using the multilevel lens, and provide insights based on the available literature rather than showing any effects of MSAs. There ere no or limited findings on specific components UHC, and the whole findings section/discussion section was framed accordingly. 

Results

Line 248-249: what do you mean to say by many NTDs-related morbidities cause tobacco and road traffic accidents. And the ref you have cited for this statement (21, 28, 36) do not seem relevant to this line.

Response: We have corrected these in the revision. 

Line 278: in one of the other errors of wrong (?) citation where you cite an information about earthquake response in Nepal from a publication from southern Australia. Please check these kinds of errors. There can be many which might go under radar of editor or reviewers. By taking a complete responsibility of ensuring a valid write up with proper citation, I suggest a complete run-through and thorough review of all citations and correcting any errors that the paper contains. This is also highlighted by one of the reviewers.

Response: we again thank you for picking up these errors. We rechecked all references and made necessary corrections accordingly.

Line 285 and para following it. You talk about community health organizations and have mixed that with committees for MSAs. However, the community level activities fall under micro level levers as per your framework. I suggest not mixing up the community engagement even if it is through participation in the local health facility committees. Please delineate that from the meso level committees and put it in micro level.

Response: thank you for this feedback; we have revised as suggested in the revised manuscript.

Reaching towards the end of the results section, I was expecting some key findings on how you show linkages of multisectoral actions PHC and UHC. I could see something on PHC but found that UHC is completely missing. I struggled to note what exactly you found about UHC and which dimension of UHC is supported with this evidence.

Response: As explained in the response in the earlier comment, the role of MSAs in PHC has multiple pathways. For example, multisectoral actions at the macro-level act as strategic inputs through macrolevel multisectoral committees, working groups, steering groups, and secretariat (producing joint actions plan, multisectoral action plan) where there is little explanation of UHC. However, they act as inputs for the PHC. At the meso level (where macro-policies and plans are operationalized), multisector come together to try to mitigate and address the effects (here, MSAs address the spillover effects through sectoral offices of the line ministries, for example in response to earthquake impacts, multiple sectors work together including water and sanitation office, supplies of foods, communication, transport. At the meso level, MSAs contribute to implementation of PHC; however, there was limited discussion of UHC. However, these policies and actions support implementing PHC and implicitly contribute to achieving UHC. At the micro-level, where service users and providers interface, MSAs in PHC contribute to service provision, delivery and utilization of health services (from the provider side), health service use, health promotion and prevention, and lifestyle change of populations (users’ side). As in our review findings, there were limited results on UHC. Therefore, we made minor change removing UHC from the title and focus accordingly. We revised our work focusing on MSAs and PHC, specifically with the implicit outcome of equity and universality. 

Discussion and conclusions: also, as commented by one of the reviewers, I struggled to pick a take home message from this review. I would say a lot of information and some limitations as commented above have made you struggle to come up with a clear conclusion. The objective and the research question were not found to have been supported with the conclusion.

Response: in the revised version, we added limitations as suggested. Additionally, we clarified the conclusion articulated with the study objective/research question.

Minor comments: The language part needs to be thoroughly checked. There are issues like inconsistent use of capital letters in a number of places. And the issues pointed out by one of the reviewers on use of abbreviated forms also has rampantly occurred in the paper. It looks like the authors have not given enough attention to review their final version of the paper. Please correct all the language issues including these.

Response: thank you for the suggestions; we corrected errors throughout.

At the end, I would suggest that the authors take a serious look on their aim of what they expected to depict from this review, revise the methods section as mentioned above and also other sections. The paper in the current form does not match the journal’s standard without a major revision.

Response: thank you for the suggestions we corrected errors throughout.

Reviewer #1: I congratulate the authors for concluding their scientific efforts logically, but I would recommend them to consider the following aspects while finalizing the manuscript.

1. Title of the manuscript: I don't think the word "multilevel" is required in the title. I would suggest removing this word from the title. As Principles of PHC has spelled- "Multisectoral Coordination" not the "Multisectoral action". I am afraid, multisectoral action may misguide the readers.

Response: thank you for the suggestions. Since 1970s, multisectoral coordination was used, while the recent literature referred to, multisectoral coordination was used, while the recent literature referred to MSAs for the same terminology. Therefore, we used to MSAs to denote intersectoral coordination. As the core framing of this paper contribution of MSAs to PHC at the multilevel health systems (that improve equity and universality). We removed the multilevel from the title, as suggested.

2. The authors are presenting the findings which are more relevant to LMICs/Limited resources settings which may mislead the readers that 'multisectoral coordination' (? Actions) is not relevant in high income countries/resource abundant settings.

Response: We agree with the reviewer that the ideas of MSAs are more relevant to resource-limited settings. However, some studies (9 studies) from high income countries reported that MSAs are important in these settings too. 

3. While reviewing the manuscript, I tried to pick a take home message that could be useful for health system workers/decision makers, but I found no new idea/message from the manuscript. Multisectoral coordination had been identified as one of the core principles of PHC since 1978. What new idea/knowledge this study adds in the PHC science? Did the authors just attempt to use the RAMESES standards/check list in the area of PHC. I would suggest authors to highlight on the key but nobel idea/knowledge.

Response: Generally, review studies (like literature reviews and scoping reviews) synthesize existing evidence and provide insights/perspectives from the available data. In other words, explain and interpret the findings based on available evidence. In this scoping review, we synthesized the evidence, explained them using a multilevel framework, and provided how they operated at the macro, meso, and micro-level health systems contributing to the policy, operation and delivery of PHC to achieve desired health system goal (UHC- equity, and universality). We have incorporated these insights in the paper, including the conclusion of the paper.

Reviewer #2: Dear Authors, Thank you for your hard work and intelligent piece of the write up to bring forward theoretically guided scoping review for the multilevel and multisectoral actions in primary health care. The interesting finding to highlight the spillover effect of the non-health sector actors is appreciated.

Response: thank you for this appreciation. We appreciate your feedback and encouragement. 

Please find some feedback for your consideration:

- Introduction: Line 66 better to mention1978; can combine line 69 and 73; line 81-86 secondary citation is used, better to quote primary references.

Response: thank you for this; we made corrections for this.

- The search strategy includes Universal Health Coverage, the results and discussion section do not much highlight on the inter-relation of the delivery of PHC to achieve UHC; fails to give clear picture to the reader. This linkage is also missing in the introduction part. Justification added to why UHC was used in the search strategy and linking it up to the rest of the article is needed. Suggested title "Multilevel multisectoral policy and actions in primary health care for achieving universal health coverage: a realist synthesis of scoping review"

Response: thank you for this very insightful comment. As overall framing of the paper was to describe the context/mechanisms/ pathways of MSAs on the design and implementation of PHC towards equity/universality. To capture the records related to equity/universality (in terms of access, coverage, and quality) of health services, we included the search terms as UHC (instead to focus on three dimensions of UHC). As in our review findings, our focus was not to dig down the UHC (in terms of service, population, or financial protection), we removed the UHC form title and framed the whole paper accordingly. We revised our work focusing on MSAs and PHC specifically with implicit outcomes of equity and universality. As per the suggestion, we also changed the manuscript's title.

- Exclusion criteria are not clear. Line 161/162 reads "sufficient information"; what do this mean is not clear, please mention concretely what it meant to the authors so as to bring readers on the same page and add these to the flow diagram as reasons of exclusion.

Response: thanks for this feedback. We have added the reasons for exclusion.

-The citation though has article on multisectoral actions for NCDs (64), very less is described in the result and discussion on aspects of macro and meso level from NCDs lens; case studies on Multisectoral action plan for prevention and control of NCDs for any country (example:https://www.who.int/docs/default-source/nepal-documents/multisectoral-action-plan-for-prevention-and-control-of-ncds-(2014-2020).pdf?sfvrsn=c3fa147c_4 ) is suggested. In addition AMR is a known multisectoral agenda; suggestion is to review on case study on AMR for the spillover effect (example: https://www.hindawi.com/journals/jtm/2022/2783300/). These literatures will work to strengthen the comprehensiveness of the review and the conclusion.

Response: thanks for this feedback. We have cited these references in the revised manuscript.

- In supplementary table, health literacy has come up as one of the pertinent findings; it is one of the important factors acknowledged for lifestyle changes and behavior modification; suggested to bring health literacy to the study results, discussion and conclusion too.

Response: thanks for this feedback. We have revised it as suggested.

- Commercial determinants of health has been kept in the supplementary table, however it has not been taken up in the main write up. Suggestion to include it to integrate the list in the discussion that is fitting into commercial determinants of health (https://www.thelancet.com/series/commercial-determinants-health)

Response: thanks for this feedback. We have added the reasons for exclusion.

- Since it is scoping review, it will have its certain limitations: Add limitation of your study

Response: thanks for this feedback. We have added more limitations of the study.

Language impression: There are instances of use of the short forms without their full forms and inconsistent short forms used; eg- Line 91 UHC, Line 235 SDH- SoDH elsewhere; Line 355 NCDs and so on. Suggestion to use short form followed by full form for the first time use and then consistent to use the same short form thereafter.

Response: thanks for this feedback. We have corrected as suggested.

Thank you for the excellent work.

Response: thank you so much for appreciation.

---

## [Decision Letter · Decision Letter 1]

19 Jul 2023

PONE-D-23-09012R1Multisectoral actions in primary health care: a realist review of evidencePLOS ONE

Dear Dr. Khatri,

Thank you for submitting your manuscript to PLOS ONE. After careful consideration, we feel that it has merit but does not fully meet PLOS ONE’s publication criteria as it currently stands. Therefore, we invite you to submit a revised version of the manuscript that addresses the points raised during the review process.

Few minor comments below. 

We look forward to receiving your revised manuscript.

Kind regards,

Krishna Kumar Aryal

Academic Editor

PLOS ONE

Journal Requirements:

**Additional Editor Comments:**

Title – the changes on the last part of the title to me did not look great. Up to the authors but the previous line saying a realist synthesis of scoping review looked to be a better presentation of the work rather than the new one changed in the revised version.

Intro –In the starting line of the introduction, if you could interweave intersectoral coordination on health referring to alma ata into what you have written as MSA being fundamental principle of PHC, it would make this read even more beautiful. In line 98, the use of sentence We employed….. gives a sense of methods section being injected in the intro. Authors might want to reconsider if they really want to position any methods language in the intro.

One more serious and thorough copyediting required. Abbreviations – there still remain issues. Like MSA coined early in the abstract but later in couple of places again full form is still there (and in few places in the main body of the manuscript after it has been abbreviated in the beginning). It is suggested to take this manuscript through a thorough copyediting to make sure these kind of errors as well as errors like inconsistent use of capital letters to name a few are corrected including all other typographical issues. Some more examples of typographical issues (there could be more) Line 106 - …different categories MSAs on health…. Is something missing here? Probably ‘of’? Line 470 – did the authors mean multilevel context of health systems and not health and systems.

Reviewers' comments:

Reviewer's Responses to Questions

**Comments to the Author**

1. If the authors have adequately addressed your comments raised in a previous round of review and you feel that this manuscript is now acceptable for publication, you may indicate that here to bypass the “Comments to the Author” section, enter your conflict of interest statement in the “Confidential to Editor” section, and submit your "Accept" recommendation.

Reviewer #1: All comments have been addressed

Reviewer #2: All comments have been addressed

2. Is the manuscript technically sound, and do the data support the conclusions?

Reviewer #1: Yes

Reviewer #2: Yes

3. Has the statistical analysis been performed appropriately and rigorously? 

Reviewer #1: N/A

Reviewer #2: N/A

4. Have the authors made all data underlying the findings in their manuscript fully available?

Reviewer #1: Yes

Reviewer #2: Yes

5. Is the manuscript presented in an intelligible fashion and written in standard English?

Reviewer #1: Yes

Reviewer #2: Yes

6. Review Comments to the Author

Reviewer #1: Dear Authors

Congratulation for developing the manuscript in the most relevant issue of global health.

Despite your rigorous analysis on the study subject, still we expect more analytical and crucial findings.

Once again, thank you for addressing the comments raised during the first review process.

Reviewer #2: (No Response)

7. PLOS authors have the option to publish the peer review history of their article (what does this mean?). If published, this will include your full peer review and any attached files.

Reviewer #1: **Yes: **Bhim Prasad Sapkota

Reviewer #2: **Yes: **Ambika Thapa

---

## [Author Response · Author response to Decision Letter 1]

25 Jul 2023

Point by point to the editor's and reviewers’ comments 

The authors’ team would like to thank you for constructive feedback on our manuscript. We appreciate your insightful and constructive feedback. We fully agree with your views. We have revised as suggested. In this document, we have responded point-by-point on your feedback, and clarification of the concerns where necessary. 

Additional Editor Comments:

Comments Response 

Title – the changes on the last part of the title to me did not look great. Up to the authors but the previous line saying a realist synthesis of scoping review looked to be a better presentation of the work rather than the new one changed in the revised version. Corrected as suggested. 

Intro –In the starting line of the introduction, if you could interweave intersectoral coordination on health referring to alma ata into what you have written as MSA being fundamental principle of PHC, it would make this read even more beautiful. 

In line 98, the use of sentence We employed….. gives a sense of methods section being injected in the intro. Authors might want to reconsider if they really want to position any methods language in the intro. Thank you for this feedback. We revised the concept of intersectional coordination in 1978. We hope this make sense to the readers.

We revised the method-related contents from the introduction section. 

One more serious and thorough copyediting required. Abbreviations – there still remain issues. Like MSA coined early in the abstract but later in couple of places again full form is still there (and in few places in the main body of the manuscript after it has been abbreviated in the beginning). 

It is suggested to take this manuscript through a thorough copyediting to make sure these kind of errors as well as errors like inconsistent use of capital letters to name a few are corrected including all other typographical issues. Some more examples of typographical issues (there could be more) Line 106 - …different categories MSAs on health…. Is something missing here? Probably ‘of’? Line 470 – did the authors mean multilevel context of health systems and not health and systems. We checked thoroughly these issues and corrected. 

We edited our manuscript throughout. Thank you so much for suggestions.

Thank you so much for so constructive feedback on our work.

---

## [Editor Report · Decision Letter 2]

27 Jul 2023

Multisectoral actions in primary health care: a realist synthesis of scoping review

PONE-D-23-09012R2

Dear Dr. Khatri,

We’re pleased to inform you that your manuscript has been judged scientifically suitable for publication and will be formally accepted for publication once it meets all outstanding technical requirements.

Kind regards,

Krishna Kumar Aryal

Academic Editor

PLOS ONE

Additional Editor Comments (optional):

Thank you for addressing the remaining issues. A good paper in the current context of high need but minimum action on intersectoral or multisectoral coordination.
---

## [Editor Report · Acceptance letter]

31 Jul 2023

PONE-D-23-09012R2 

Multisectoral actions in primary health care: a realist synthesis of scoping review 

Dear Dr. Khatri:

I'm pleased to inform you that your manuscript has been deemed suitable for publication in PLOS ONE. Congratulations! Your manuscript is now with our production department. 

Kind regards, 

on behalf of

Dr. Krishna Kumar Aryal 

Academic Editor

PLOS ONE